# Prototype Regularized Manifold Regularization Technique for Semi-Supervised Online Extreme Learning Machine

**DOI:** 10.3390/s22093113

**Published:** 2022-04-19

**Authors:** Muhammad Zafran Muhammad Zaly Shah, Anazida Zainal, Fuad A. Ghaleb, Abdulrahman Al-Qarafi, Faisal Saeed

**Affiliations:** 1School of Computing, Faculty of Engineering, Universiti Teknologi Malaysia, Iskandar Puteri 81310, Malaysia; anazida@utm.my (A.Z.); abdulgaleel@utm.my (F.A.G.); 2College of Computer Science and Engineering, Taibah University, Medina 42353, Saudi Arabia; asalqarafi@taibahu.edu.sa; 3Data Analytics and Artificial Intelligence (DAAI) Research Group, Department of Computing and Data Science, School of Computing and Digital Technology, Birmingham City University, Birmingham B4 7XG, UK; faisal.saeed@bcu.ac.uk

**Keywords:** machine learning, semi-supervised learning, manifold regularization, sequential learning, Internet of Things

## Abstract

Data streaming applications such as the Internet of Things (IoT) require processing or predicting from sequential data from various sensors. However, most of the data are unlabeled, making applying fully supervised learning algorithms impossible. The online manifold regularization approach allows sequential learning from partially labeled data, which is useful for sequential learning in environments with scarcely labeled data. Unfortunately, the manifold regularization technique does not work out of the box as it requires determining the radial basis function (RBF) kernel width parameter. The RBF kernel width parameter directly impacts the performance as it is used to inform the model to which class each piece of data most likely belongs. The width parameter is often determined off-line via hyperparameter search, where a vast amount of labeled data is required. Therefore, it limits its utility in applications where it is difficult to collect a great deal of labeled data, such as data stream mining. To address this issue, we proposed eliminating the RBF kernel from the manifold regularization technique altogether by combining the manifold regularization technique with a prototype learning method, which uses a finite set of prototypes to approximate the entire data set. Compared to other manifold regularization approaches, this approach instead queries the prototype-based learner to find the most similar samples for each sample instead of relying on the RBF kernel. Thus, it no longer necessitates the RBF kernel, which improves its practicality. The proposed approach can learn faster and achieve a higher classification performance than other manifold regularization techniques based on experiments on benchmark data sets. Results showed that the proposed approach can perform well even without using the RBF kernel, which improves the practicality of manifold regularization techniques for semi-supervised learning.

## 1. Introduction

An application such as the Internet of Things (IoT) generates large amounts of data constantly that needs to be processed. Due to the large amount of data constantly generated via various sources such as sensors and mobile devices, it becomes increasingly impractical for batch learning algorithms that require a full data set to fit in the memory for learning [1]. Solving this problem requires a more efficient learning algorithm that processes data sequentially without requiring the whole data set to be stored in the memory for learning to commence [2]. By processing data sequentially, each piece of data can be deleted from the memory once it is processed, thus reducing memory consumption. However, the current online learning algorithms are supervised learning algorithms that supposedly assume the whole data set or data stream is labeled [2]. The constant availability of labeled data is a flawed assumption as labeling each piece of data for large data sets or in a streaming environment is expensive and time consuming, making the current online learning algorithms impractical for real-world applications [3]. Thus, learning algorithms that require less demand on labeled data should be designed to improve the practicality of learning algorithms for real-world applications.

An alternative approach to supervised learning is the semi-supervised learning (SSL) approach, where the learning algorithm can exploit assumptions of the data to learn from a few pieces of labeled data [4]. The most common assumptions imposed on the data in SSL are the cluster, smoothness, or manifold assumptions. For example, the cluster assumption assumes that data grouped in the same cluster would tend to have the same labels. Meanwhile, the smoothness assumption assumes that similar data in a high-density region would have similar labels [5]. However, the cluster and smoothness assumptions may be ineffective on high-dimensional data sets due to the curse of dimensionality [6]. This manifests in what is known as the concentration phenomenon that causes most pairwise distances between two data points in the whole data set to be almost similar [7].

The manifold assumption may be preferred for higher-dimensional data sets as it assumes that high-dimensional data sets may be embedded in a lower-dimensional subspace [4]. SSL approaches apply the manifold assumption through the manifold regularization approach that penalizes the model during training if the prediction difference between two similar points is high [8]. This is enforced by the aptly named manifold regularization term calculated using the Laplacian graph that models the local neighborhood of the data set and embeds it in a lower dimension.

The current online approaches that apply the manifold regularization method can be divided into the Support Vector Machine (SVM) [9] approach and extreme learning machine (ELM) approach [10]. These algorithms are the most suitable for manifold regularization as they simply add the manifold regularization term to the objective function in addition to other terms such as the regularizing term to avoid overfitting and the prediction error term. Despite this, the ELM approach is easier and more practical to implement than SVM as ELM has a closed-form solution [11]. In contrast, the SVM approach requires an online gradient descent to optimize its parameters, which may not converge to the global optimum [12].

However, applying the manifold regularization method in real-world environments is still lacking practicality. This is due to the difficulty in determining the width parameter λ of the radial basis function (RBF) kernel in Equation (1). The width parameter is important to describe the model in which data are similar and are most likely to belong to the same class. The importance of the width parameter γ is illustrated in Figure 1, where the RBF kernel is centered at x=(1.0,−0.5) with the brighter colors indicating the high assignment of values by the RBF kernel and the darker colors indicating the lower assignment by the RBF kernel. In Figure 1a, the parameter γ=0.1 is poorly selected as the mass of the RBF kernel covers the whole data set and is uninformative as it indicates that all data belong to the same class. Figure 1b shows a better parameter selection of γ=1 as the mass of the RBF kernel is mostly located in the same class but touches slightly to the opposite class. On the other hand, Figure 1c only covers the same class but is too narrow to only the data surrounding the center, making learning slow. Finally, an optimal width parameter for the RBF kernel would be γ=4, as shown in Figure 1d, as it covers a wide enough region in the data set but the mass focuses on data in the same class.
(1)$$k\left(x_{i},x_{i}\right)=\exp\left(-\gamma \|x_{i}-x_{j}\|{}^{2}\right)$$

Current approaches determine this parameter by using hyperparameter selection techniques such as grid search or relying on previous experience with the data set [9,13]. Hyperparameter selection techniques require many pieces of labeled data to provide accurate performance estimates for each parameter selection. This is counterproductive to be applied in environments where a large amount of labeled data is difficult to be obtained. Furthermore, in applications such as data stream mining, it is impossible to collect and store all the data in the first place due to the possibly infinite arrival of data, thus making determining the correct RBF kernel width parameter γ difficult in data stream settings.

To address this issue, we proposed combining a prototype-based learning algorithm with the manifold regularization technique to preserve the local neighborhood of each piece of data. The prototype-based learning algorithm is a technique that approximates the data set via a set of prototypes to extract information from the data, such as to detect clusters, which can be used to classify new data. Some examples of prototype-based learning algorithms are the self-organizing neural network (SOINN) [14], online semi-supervised-learning vector quantization (OSS-LVQ) [12], and Gaussian mixture models (GMM) [15]. Prototype-based learning algorithms have had successes in applications such as image classification [16,17] and vibration diagnosis [18].

Our study aimed to propose combining a prototype-based learning algorithm, specifically the enhanced self-organizing incremental neural network (ESOINN) [19] with the semi-supervised online sequential-extreme learning machine (SOS-ELM) [20] manifold regularization approach, which this proposed approach will be built upon. We chose SOS-ELM as the basis of this approach due to its sequential data processing characteristic and efficient updating procedure that consumes less memory compared with the SVM-based approaches. This makes the SOS-ELM suitable for applications in data stream applications such as IoT.

Essentially, the ESOINN will store the prototypes that will be used for the SOS-ELM to select its nearest neighbors to construct the Laplacian graph. Therefore, compared to other manifold regularization approaches, this approach does not simply use the new incoming data to construct the Laplacian graph but only uses the prototypes in the ESOINN. This eliminates the requirement of using the RBF kernel and determining the width parameter to specify which data are similar to the center. This is because each piece of data is always regularized with data, i.e., the prototypes that are in its local neighborhood. Additionally, we modified the ESOINN to take advantage of the labeled data to improve the separation between classes to ensure that only prototypes from similar classes are selected as the nearest neighbor for each sample.

Through experiments, this proposed approach can learn faster by using the prototypes to construct the Laplacian graph. This shows that the manifold assumption, which states that high-dimensional data are locally Euclidean, is verified. Moreover, this approach also shows significant performance improvement on an imbalanced data set. However, this comes at a slight increase in computational complexity to train the modified ESOINN. The contributions of this study are summarized below:(1)A prototype regularized manifold regularization approach that eliminates the requirement of the RBF kernel, which in turn no longer necessitates the width parameter γ to be determined for optimum performance.(2)A modified ESOINN that utilizes the labeled data to improve the cluster separation between different classes. This helps the prototype manifold regularization approach in (1) to only regularize to samples from the same class, which strengthens the smoothness assumption of SSL.

The rest of this paper is organized as follows. In Section 2, recent works that were conducted related to this study are discussed. Next, in Section 3, the relevant background information is described. In Section 4, the proposed prototype regularized manifold regularization is presented. Then, Section 5 describes the experimental setup and presents the results of the experiments. Finally, in Section 6, the key findings of this study are pointed out to conclude this paper.

## 2. Literature Review

### 2.1. Manifold Regularization Semi-Supervised Learning

The manifold regularization method in semi-supervised learning (SSL) utilizes the smoothness assumption in which it assumes that similar data tend to have similar labels [8]. In this approach, the objective function is designed to penalize the model if it makes the wrong prediction based on the labeled data and penalizes large prediction differences on similar data via the manifold regularization term [21]. Prediction algorithms that apply this approach are the support vector machine (SVM) and extreme learning machine (ELM) algorithms. The manifold regularization term could simply be added to the loss function to train these algorithms.

For SVM, the Laplacian-SVM (LapSVM) [9] was the first SVM that utilized manifold regularization to learn from unlabeled data. LapSVM simultaneously learned the max-margin classifier to separate the data set into two classes and minimized the prediction distance between similar data. However, LapSVM is ineffective for large data sets as the computation of the optimization grows cubically $O\left(n^{3}\right)$ with the number of samples [22], making it impossible for large data sets to fit in the memory. Therefore, the focus of the research on manifold regularization of SVM was directed to improving the computational efficiency to train the SVM.

Attempting to learn the LapSVM sequentially to update the weights only reduces the computational complexity to $O\;\left(n^{2}\right)$ as all samples would still need to be stored and will exhaust all the available memory resources in a streaming environment [23]. The distributed online S^3^VM (dS^3^VM) [24] is an online training approach proposed to compute the manifold regularization term using samples of the data set instead of the whole data set. The dS^3^VM approach achieves this by selecting several anchor points that act as the distribution center for the similarity matrix to be calculated between new data. Another approach, which is similarly called distributed S^3^VM [25], uses a hardware-oriented approach that solves the optimization problem across multiple processors distributed in a networked environment. However, these approaches only converge to the global optimum asymptotically, and no theoretical convergence rate is provided [12,24].

The ELM-based approach is relatively straightforward as its optimization procedure is derived from the least square approximation procedure, which is guaranteed to converge to the global optimum and has a closed-form solution. The semi-supervised-ELM (SS-ELM) [10] was the first ELM that applied the manifold regularization method. Similar to the LapSVM, this, too, is impractical for large data sets. However, for this approach, it is due to the cost of computing the pseudo-inverse to solve the closed-form solution [20]. Therefore, an online learning algorithm that processes data sequentially is preferred compared to the offline SS-ELM to avoid computing a large matrix inverse. However, compared to the SVM approach mentioned earlier, online manifold regularization ELM does not require the whole data set at each iteration by default.

The semi-supervised online sequential-ELM (SOS-ELM) [20] is an online version of the SS-ELM that is derived from the recursive least square (RLS) algorithm to update the parameters iteratively. The incremental Laplacian regularized-ELM (ILR-ELM) [26] improves the SOS-ELM by being able to update the parameters even when no labeled data are available in the most recent chunk. The ILR-ELM improves the practicality as the labeled data are not required at each iteration. Finally, the elastic SOS-ELM (ESOS-ELM) [27] can regulate the trade-off between the importance of learning from labeled and unlabeled data to allow dynamic hyperparameter tuning.

However, these approaches do not preserve the local neighborhood of samples in each chunk as they only use samples from the current chunk to construct the regularization term. As a result, some samples may be paired with samples distant from each other, which does not obey the smoothness assumption. Furthermore, manifold regularization techniques require determining the RBF kernel width parameter, which is difficult to determine in a fully online setting and without the availability of sufficiently excess labeled data. In this approach, we proposed a prototype-based approach to construct the regularization term instead of relying on the RBF kernel.

### 2.2. Prototype-Based Learning Methods

Prototype learning methods aim to learn a set of prototypes that can approximate an entire data set [12]. A popular prototype-based learning method is the K-means algorithm, which uses a set of n observations to partition the data set into k clusters [28]. However, the K-means algorithm may produce poor clusters on oddly shaped data sets, such as in the moons’ data set, as it can only work on circular-shaped clusters.

Another more robust prototype learning method that can detect odd-shaped clusters is the topology learning method, which learns the structure of data via a set of nodes interconnected via edges. Examples of these algorithms are the growing neural gas (GNG) [29], self-organizing map (SOM) [30], and the self-organizing incremental neural network (SOINN) [31]. The GNG algorithm uses Hebb’s learning rule to sequentially update the nodes by pulling the two closest nodes to the newly observed data. A new node will also be added between nodes with the highest accumulated error to increase the capacity of the model to learn and approximate the data set. Meanwhile, SOM learns the topology, similarly to the GNG algorithm, but updates the entire node set weighted by the distance to the data, i.e., the closer the data, the larger the update, and vice versa.

Both the GNG and SOM are capable of learning incrementally, which suits applications such as data stream mining. However, in data stream mining, there may often be noise-generated data that may result from noisy sensors and may cause both GNG and SOM to learn from these noisy data accidentally. A more robust topology learning method is the SOINN method, which can filter out noise-generated nodes by deleting isolated nodes that have no edges connected to them. An improvement to the SOINN algorithm is the enhanced SOINN (ESOINN) [19] algorithm that filters out nodes with fewer than two connected nodes and also with a winning time that is significantly less than other nodes in the network. However, prototype-based learning algorithms are currently limited to unsupervised learning tasks to uncover structures in the data set instead of building a classification function as it applies lazy classification algorithms, e.g., KNN, which does not scale well on high-dimensional data sets [7].

## 3. Background

### 3.1. Enhanced Self-Organizing Incremental Neural Network

The enhanced self-organizing incremental neural network (ESOINN) [19] is derived from the SOINN algorithm [31] that discovers clusters of data by learning the topology of the data by connecting similar data represented by a node (or prototype) via an edge. This results in ESOINN having a network-like structure that resembles an undirected graph where each node represents a data point and each edge represents connections between data points.

ESOINN is trained by incrementally adding new nodes or connecting two existing nodes depending on the novelty of the new input signal. The novelty criterion is determined by comparing the similarity of the new node with the nearest node (winner) $N_{1}$ and the second nearest node (second winner) $N_{2},$ as in the condition in line 8 of Algorithm 1. $T_{1}$ and $T_{2}$ are the similarity thresholds obtained via Equation (2) using the Euclidean distance as the distance measure. A new node is considered new if it is dissimilar enough to both the first nearest node and the second nearest node. Henceforth, a new node will be added if it satisfies the condition.
(2)$$T_{i}=\{\begin{array}{l} \underset{a\in N_{i}}{\min}\|N_{i}-a\|,\;N_{i}\neq \emptyset \\ \underset{a\in S/N_{i}}{\max}\|N_{i}-a\|,\;N_{i}=\emptyset \end{array}$$

**Algorithm 1.** Enhanced Self-Organizing Incremental Neural Network (ESOINN) **Input:** Data set $D={\left\{x_{i}\right\}}_{i=1}^{N}$, maximum egde age $age_{max}$, noise remove interval λ
**Output:** Set of prototypes S
1: **function** train_esoinn ($D,age_{max},\lambda$):2: **foreach**
x∈D:3:        **if**
|S|
**<** 2:4:                S=S∪x
5:                **continue**6:        $N_{1}=\underset{a\in S}{\min}\|a-x\|$ //find 1st winner7:        $N_{2}=\underset{a\in S\backslash N_{1}}{\min}\|a-x\|$ //find 2nd winner8:        **if**
$\|x-N_{1}\|>T_{1}$
**or**
$\|x-N_{2}\|>T_{2}$:9:              S=S∪x
10:        **else:**11:                $N_{1}=N_{1}-\frac{\left(x-N_{1}\right)}{WT\left(N_{1}\right)}$ //update 1st winner12:                $a=a-\frac{\left(x-N_{1}\right)}{100\cdot WT\left(N_{1}\right)},\;a\in N_{1}$ //update 1st winner neighbors13:                $WT\left(N_{1}\right)=WT\left(N_{1}\right)+1$
14:                **if**
$e\left(N_{1},N_{2}\right)\notin ℰ$:15:                $ℰ=ℰ\cup e\left(N_{1},N_{2}\right)$ //add a connection between 1st winner and 2nd winner16:                **else:**17:                              $age\left(N_{1},N_{1}\right)=0$
18:                $age\left(N_{1},a\right)=age\left(N_{1},a\right)+1,\;a\in N_{1}$ //increment ages of edge of 1st winner neighbors 19:               remove all edges if age > $age_{max}$
20:               **if** number of data is multiple of λ:21:                     **foreach**
a∈S:22:                            **if |**$N_{a}|=0:$
23:                                    S=S\a //remove node a24:                            **else if**
|Na|=1 **and**
WT(a)<C1·WTmean:25:                                             S=S\a //remove node a26:                            **else if**
|Na|=2 **and**
WT(a)<C2·WTmean:27: **end function**

If the new signal is not considered novel, it provides evidence that the winner and second winner belong to the same cluster as both of them are similar. Therefore, an edge is added between the first and second winners if no connection exists between them. If an edge already exists, the edge age is reset to zero and the winner and its neighbors are merged closer to the input signal using the update formula given by Equations (3) and (4). The edge emanating from the winner node $N_{1}$ is then incremented by one.



(3)
$$N_{1}=N_{1}-\frac{1}{WT\left(N_{1}\right)}\left(x-N_{1}\right)$$


(4)
$$a=a-\frac{1}{100WT\left(N_{1}\right)}\left(x-a\right),\;a\in N_{1}$$



When a user-defined interval λ is reached, the node pruning procedure is executed to remove possible noisy nodes that may be accidentally added during the interval. In the original SOINN algorithm, noisy nodes are marked by nodes that are not connected to any node. ESOINN extends this condition to allow connected nodes to be marked as noisy nodes. In this case, a node is considered noise if it has two or fewer edges and its winning time WT(a) is significantly less than the average winning time in the network.

### 3.2. Overview of Extreme Learning Machines and Semi-Supervised Extreme Learning Machines

An extreme learning machine (ELM) [32] is a single-layer feedforward neural network (SLFN), which is a neural network that consists of only a single hidden layer. Despite having only a single hidden layer, ELM has been proven to have the universal approximation property that allows it to approximate any real-valued function given a finite number of hidden layer nodes [33]. However, in practice, a stable performance could be achieved by using 1000 hidden nodes [32,34].

The prediction (output) $\hat{T}$ of an ELM is given by Equation (5), where H is the input-hidden matrix and β is the hidden-output weight. The input-hidden matrix H (Equation (6)) is parameterized by the activation function $G\left(a_{i},b_{i},x_{i}\right)$ where $a_{i}$ and $b_{i}$ are the input and bias weight, respectively, that are assigned randomly via the normal distribution.
(5)$$\hat{T}=H\beta$$
(6)$$H={\left[\begin{array}{ccc} G\left(a_{1},b_{1},x_{1}\right) & \cdots & G\left(a_{L},b_{L},x_{1}\right)\\ \vdots & \ddots & \vdots \\ G\left(a_{1},b_{1},x_{N}\right) & \cdots & G\left(a_{L},b_{L},x_{N}\right) \end{array}\right]}_{N\times L}$$

Training an ELM is achieved by obtaining the hidden-output weights β that minimize the objective function (Equation (7)) where T is the true label given a training set $D={\left\{\left(x_{i},t_{i}\right)\right\}}_{i=0}^{N}$ of size N. By the least square approximation, the hidden-output weights β that minimize the objective function (Equation (7)) are given by Equation (8), where $H^{†}$ is the Moore–Penrose pseudoinverse of the input-hidden matrix H.
(7)$$\underset{\beta }{\min}\|H\beta -T\|$$
(8)$$\beta =H^{†}T.\;$$

However, the ELM has two key limitations. The ELM assumes labeled data are always available and that it is impractical for large data sets due to the cubic cost of $O\left(n^{3}\right)$ to compute the pseudoinverse $H^{†}$. To address the labeling issue, the semi-supervised ELM [14] was introduced using the manifold regularization method that utilizes the smoothness assumption of SSL. The smoothness assumption assumes that if both pieces of data are similar, they tend to have similar labels.

The smoothness assumption is enforced by the manifold regularizing term (Equation (9)), which penalizes the divergence of the prediction $\left\|{\hat{t}}_{i}-{\hat{t}}_{j}\right\|$ if the data are similar, given by the similarity matrix $W=\left[w_{ij}\right]$. The similarity matrix W is sparse by only populating the k-nearest neighbors of each data, and the weights are given by the RBF kernel $\exp\left(-\lambda \|x_{i}-x_{i}\|{}^{2}\right)$, where λ is the width parameter or fixed to 1. The calculation of the regularizing term (Equation (9)) can be simplified using a more compact function (Equation (10)), where L is the Laplacian graph matrix L=D−W and D is the degree matrix with elements $D_{ii}=\overset{N}{\underset{j=0}{\sum }}w_{ij}$
(9)$$L_{m}=\frac{1}{2}\underset{i,j}{\sum }w_{ij}\|{\hat{t}}_{i}-{\hat{t}}_{j}\|$$
(10)$$L_{m}=\mathrm{Tr}\left({\hat{T}}^{T}L\hat{T}\right)$$

The term $L_{m}$ can then be used as a regularizing term of the ELM in the new objective function (Equation (11)), where $J=diag\left(C_{i},\ldots ,C_{l},0,\ldots ,0\right)$ with its first l labeled data elements equal to C to control overfitting and zero everywhere else and α controls the trade-off between the importance of labeled and unlabeled data. Again, by the least square approximation technique, the parameter β that minimizes the objective (Equation (11)) can be obtained via the closed-form solution given by Equation (13).
(11)$$\underset{\beta }{\min}\frac{1}{2}{\left\|\beta \right\|}^{2}+{\left\|J\hat{T}-T\right\|}^{2}+\alpha {\hat{T}}^{T}L\hat{T}$$
(12)$$L_{m}=\mathrm{Tr}\left({\hat{T}}^{T}L\hat{T}\right)$$
(13)$$\beta^{*}={\left(I+H^{T}H+\alpha {\hat{T}}^{T}L\hat{T}\right)}^{-1}H^{T}JH$$

The semi-supervised online sequential-ELM (SOS-ELM) [20] is a more practical method than SS-ELM by allowing the parameter β to be updated sequentially rather than requiring the whole data set, which makes it expensive to compute the matrix inverse, as mentioned earlier. Using the recursive least square approximation technique, the sequential estimation of the parameter β is given by Equations (14) and (15).
(14)$$\beta^{\left(k+1\right)}=\beta^{\left(k\right)}+P_{k}H_{k+1}^{T}\left(J_{k+1}T_{k+1}-\left(J_{k+1}+\alpha L_{k+1}\right)H_{k+1}\;\beta^{\left(k\right)}\right)$$
(15)$$P_{k+1}=P_{k}-P_{k}H_{k+1}^{T}{\left(I+\left(J_{k+1}+\alpha L_{k+1}\right)H_{k+1}P_{k}H_{k+1}^{T}\right)}^{-1}.\left(J_{k+1}+\alpha L_{k+1}\right)H_{k+1}P_{k}$$

The pseudocode to update and train the SOS-ELM is described below.
**Initialization Phase:**Obtain an initial data set $D_{o}=\left\{\left(x_{1},y_{1}\right),\ldots ,\left(x_{N_{o}},\;y_{N_{o}}\right)\right\}$ with true labels1. Randomly assign input weights $a_{i}$ and bias $b_{i}$ for i=1,…, L2. Calculate the hidden layer output matrix:3. Calculate the initial hidden-output weights:(16)$$\beta^{o}=P_{o}H_{o}^{T}Y_{o}$$where(17)$$P_{o}={\left(H_{o}^{T}H_{o}\right)}^{-1},\;Y_{o}={\left(y_{1},\ldots ,y_{N_{o}}\right)}^{T}$$4. Set t=0**Sequential Learning phase:**Obtain a chunk data $D_{t}={\left\{\left(x_{1},y_{1}\right),\ldots ,\left(x_{l},\;y_{l}\right),\;\left(x_{l+1}\right),\ldots ,\;\left(x_{u}\right)\right\}}_{i=1}^{l+u}$ labeled by what consists of l labeled data and u unlabeled data.5. Calculate the hidden layer output matrix: (18)$$H_{t+1}={\left[\begin{array}{ccc} G\left(a_{1},b_{1},\;x_{1}\right) & \cdots & G\left(a_{L},b_{L},\;x\right)\\ \vdots & \ddots & \vdots \\ G\left(a_{1},b_{1},\;x_{N}\right) & \cdots & G\left(a_{L},b_{L},\;x_{N}\right) \end{array}\right]}_{N\times L}$$6. Create the Laplacian matrix L=D−W and penalty matrix J7. Calculate the hidden-output weights: (19)$$\beta^{\left(k+1\right)}=\beta^{\left(k\right)}+P_{k}H_{k+1}^{T}\left(J_{k+1}T_{k+1}-\left(J_{k+1}+\alpha L_{k+1}\right)H_{k+1}\;\beta^{\left(k\right)}\right)$$8. Calculate (20)$$\begin{array}{l} P_{k+1}=P_{k}-P_{k}H_{k+1}^{T}{\left(I+\left(J_{k+1}+\alpha L_{k+1}\right)H_{k+1}P_{k}H_{k+1}^{T}\right)}^{-1}.\left(J_{k+1}+\alpha L_{k+1}\right)H_{k+1}P_{k} \end{array}$$9. Set t=t+1

## 4. Proposed Approach

As discussed in Section 1, the main limitation of the current manifold regularization approach is that it is difficult to determine the optimal RBF kernel width parameter and may not obey the smoothness assumption as the Laplacian matrix in Equation (10) is only constructed using data in the current memory. Creating the Laplacian matrix using only data in the current memory may cause each piece of data not to be paired with other data that belong to the same class. Hence, the smoothness assumption would be violated. This can happen in imbalanced data streams when too few samples in the current data chunk belong to the minority class, as illustrated in Figure 1. The samples that belong to the minority class would then be paired with samples belonging to the majority class if the number of nearest neighbors required to create the Laplacian matrix is larger than the number of samples from the minority class.

Another issue is that the current approaches that apply the manifold regularization technique may not be robust to noise as the data may be regularized to samples generated due to noise. Furthermore, one more issue to consider is the difficulty of determining the width parameter of the RBF kernel that needs to be set based on the width of the distribution, which may be difficult to estimate from a few initial pieces of data.

To address these issues, a prototype-based manifold regularization approach was proposed to only pair each piece of data to a prototype that is its local neighborhood instead of simply pairing each piece of data to another sample in the data chunk. However, storing many samples to be used as prototypes can be inefficient in terms of memory and may not be suitable for applications such as data stream mining. Therefore, we combined the manifold regularizing approach of SOS-ELM with the ESOINN, where the ESOINN stores the library of prototypes as nodes to be used by the SOS-ELM for manifold regularization. Because ESOINN can filter out noisy nodes, it could also improve the robustness of SOS-ELM in noisy data sets.

The overview of the proposed approach is provided in Figure 2, which shows that this proposed approach consists of two main components, the prototype learner (ESOINN) and the manifold regularization learner (SOS-ELM). The main function of the ESOINN is to build a library of prototypes as nodes in the network, which will then be used by the SOS-ELM to learn a classification function. At each data arrival, the SOS-ELM queries the prototypes from the ESOINN to construct the affinity matrix as input to construct the Laplacian matrix to be used as the manifold regularizer, as in Equation (11).

For the ESOINN algorithm, we made several modifications to the original algorithm to take advantage of the labeled data to improve the separation between classes and improve its ability to filter out noisy data. The labeled data were used as an anchor to ensure nodes from similar classes were grouped. Additionally, the labeled data were used to filter out noisy nodes. This approach was reliable as the labeled data were often labeled by the human expert, which were less likely to be noisy data.

The first modification made to the ESOINN algorithm was in the adding node condition where similar to the ESOINN; the signal was checked whether it was significantly novel to be added as a new node. However, in this proposed approach, we also checked whether the signal had a label attached to it. If a label was attached, the label was assigned to the node or the label was assigned to the first winner node. This simple modification allowed the ESOINN to assign labels to its node as more data arrived incrementally.

The second modification was to prevent nodes from different classes to be connected. In line 18 of Algorithm 2, before a connection between the first and second winners was made, it checked whether both had the same label or did not have any labels assigned. If either of them was true, then a connection between the first and second winners was made.
**Algorithm 2.** Modified Enhanced Self-Organizing Incremental Neural Network (ESOINN) **Input:** Labeled and Unlabeled Data set $D={\left\{\left(x_{1},y_{1}\right),\ldots ,\left(x_{l},y_{l}\right),x_{l+1},\ldots ,x_{l+u}\right\}}_{i=1}^{N=l+u}$, maximum egde age $age_{max}$, noise remove interval λ
**Output:** Set of Prototypes S
1: **function** train_esoinn ($D,\alpha ,N_{Q},N_{S}$):2: **foreach** x or (x,y)∈D:3:        **if**
|S|
**<** 2:4:                S=S∪x
5:                **continue**6:        N1=mina∈S‖a−x‖ //find 1st winner7:        N2=mina∈S\N1‖a−x‖ //find 2nd winner8:        **if**
‖x−N1‖>T1
**or**
$\|x-N_{2}\|>T_{2}$:9:                S=S∪x
10:                **if (**x,y**):**11:                        label(x)=y
12:        **else:**13:                **if (**x,y**):**14:                        $label\left(N_{1}\right)=y\;$ //assign label to the winner node15:                $N_{1}=N_{1}-\frac{\left(x-N_{1}\right)}{WT\left(N_{1}\right)}$
16:                $a=a-\frac{\left(x-N_{1}\right)}{100\cdot WT\left(N_{1}\right)},\;a\in N_{1}$. 17:                $WT\left(N_{1}\right)=WT\left(N_{1}\right)+1$
18:             //if the label of the winners are the same or either winner has no label, connect the nodes19:            **if**
$e\left(N_{1},N_{2}\right)\notin ℰ\;and\;\left(label\left(N_{1}\right)==label\left(N_{2}\right)\right)\;or\;label\left(N_{1}\right)=label\left(N_{2}\right)==\emptyset )$:20:                $\;\;\;\;\;\;\;ℰ=ℰ\cup e\left(N_{1},N_{2}\right)$
21:                **else:**22:                             $age\left(N_{1},N_{2}\right)=0$
23:             $age\left(N_{1},a\right)=age\left(N_{1},a\right)+1,\;a\in N_{1}$
24:             **for**
$e\left(x_{1},x_{2}\right)\;\in \;ℰ:$
25:                             **if**
$e\left(x_{1},x_{2}\right)$ **or**  $age\left(e\left(x_{1},x_{2}\right)\right)>age_{max}$:26:                                         $ℰ=ℰ\backslash e\left(x_{1},x_{2}\right)$
27:             **if** number of data is multiple of λ:28:                     **foreach**
a∈S:29:                            **if** |$N_{a}|=0:$
30:                                    S=S\a //remove the node31:                            //only remove the unlabeled nodes32:                            **else if**
$\left|N_{a}\right|=1$ **and**
$label\left(N_{a}\right)==\emptyset$
**and**
WT(a)<C1·WTmean:33:                                     S=S\a //remove node a34:                        **else if**
|Na|=2 **and**
label(Na)==∅
**and**
WT(a)<C2·WTmean:35:                                S=S\a//remove node a36: **end function**

The final modification was regarding edge removal and noisy node deletion. For the edge deletion, the edge was removed if it exceeded an age threshold or if the adjacent nodes had different labels and both had labels assigned. To remove noisy nodes, we attempted to avoid removing too many nodes that may have been considered important. To prevent this, we removed nodes with no adjacent labeled nodes and significantly less winning time than other nodes.

The manifold regularization for SSL via prototype regularization can be achieved by constructing the weight matrix W, as in Equation (21). Note that in the weight matrix W that the weights are zero for the block that represents the pairs of data $x_{1:N}$ and are only nonzero between pairs of data and their respective prototypes, denoted as $p_{M,}^{N}$ which denotes the $M^{th}$ prototype of the $N^{th}$ data. This means that the data are only regularized to the prototype instead of to other data.
(21)$$W=\begin{array}{c} \begin{array}{c} x_{1}\\ \vdots \\ x_{N} \end{array}\\ \begin{array}{c} p_{M}^{1}\\ \vdots \\ p_{M}^{N} \end{array} \end{array}\left[\begin{array}{cc} \begin{array}{ccc} 0 & \cdots & 0\\ \vdots & \ddots & \vdots \\ 0 & \cdots & 0 \end{array} & \begin{array}{ccc} w_{11} & \cdots & 0\\ \vdots & \ddots & \vdots \\ 0 & \cdots & w_{NM} \end{array}\\ \begin{array}{ccc} w_{11} & \cdots & 0\\ \vdots & \ddots & \vdots \\ 0 & \cdots & w_{NM} \end{array} & \begin{array}{ccc} 0 & \cdots & 0\\ \vdots & \ddots & \vdots \\ 0 & \cdots & 0 \end{array} \end{array}\right]$$

To select the prototypes, we simply queried the nodes in the modified ESOINN and used the Euclidean distance to select M prototypes for each data $x_{1:N}$. We note that in high-dimensional data sets, the Euclidean distance is vulnerable to the concentration phenomena where the distance between two points is almost similar. However, under the manifold assumption, the data can be locally Euclidean even if they are embedded in a high-dimensional space where the Euclidean distance is insufficient to describe the distance between two points [35]. Furthermore, since we only considered the local topological structure around each piece of data, the Euclidean distance was sufficient. The pseudo-code of the overall algorithm for our proposed prototype-based manifold regularization method is proposed in Algorithm 3.
**Algorithm 3.** Prototype Regularized Semi-Supervised Online Extreme Learning Machine **Input:** Labeled and Unlabeled Data set $D={\left\{\left(x_{1},y_{1}\right),\ldots ,\left(x_{l},y_{l}\right),x_{l+1},\ldots ,x_{l+u}\right\}}_{i=1}^{N=l+u}$, maximum egde age $age_{max}$, noise removes interval λ, Nearest Neighbors K, Regularization Term C, Hidden Layer Size L.1: **function** train_esoinn ($D,\alpha ,N_{Q},N_{S}$):2: **//**initialization phase3: Obtain the initial data set $D_{o}=\left\{\left(x_{1},y_{1}\right),\ldots ,\left(x_{N_{o}},\;y_{N_{o}}\right)\right\}$ for initialization4: Process $D_{o}$ using Algorithm 2 to initialization ESOINN5: Use $D_{o}$ to calculate $\beta_{0}$ and $P_{0}$ using (16) and (17) respectively6: //sequential learning phase7: **while (**True**):**8:        Obtain a partially labeled data set $D_{t}={\left\{\left(x_{1},y_{1}\right),\ldots ,\left(x_{l},\;y_{l}\right),\;\left(x_{l+1}\right),\ldots ,\;\left(x_{u}\right)\right\}}_{i=1}^{N=l+u}$:9:        Process $D_{t}$ using Algorithm 2 to update ESOINN10:        $p_{M}^{1},\ldots ,\;p_{M}^{N}=$ obtain K prototypes each for all N data sets from the nodes in ESOINN11:        Construct weight matrix W using (21)12:        Construct Laplacian matrix L=D−W and penalty matrix J
13:        Update hidden layer parameters $\beta^{\left(k+1\right)}$ using (19)14:        Update $P^{\left(k+1\right)}$ using (20)15: **end function**

## 5. Experiments

### 5.1. Experimental Setup

This proposed approach was evaluated against semi-supervised approaches, ILR-ELM, SOS-ELM, and SSOE-ELM, as mentioned in Section 2, and fully supervised approaches, OS-ELM and RSLVQ, to evaluate the deficit in performance when labeled data are not fully available. The evaluation was benchmarked using several benchmark data sets listed in Table 1, which explains the characteristics of the data sets. In Table 1, the experimental setup, such as the number of training and test samples and the number of labeled samples supplied to the semi-supervised approaches, is also described. The selected data sets contained various challenges, such as a severe class imbalance in the KDDCup 1999 data set and noisy data as in the PAMAP2 data set. We acknowledge that there existed alternative data sets with similar challenges to the data sets in Table 1. However, we selected these data sets as they were obtained from sensors from IoT devices, which was the target application for this proposed approach and most likely contained the challenges a data stream application such as IoT might encounter. The algorithms were trained sequentially for each data set, with a chunk of 100 samples provided at each iteration. Of those 100 samples, only five samples were labeled. Hence, the total number of labeled samples was the number of training samples multiplied by five.

We adopted the interleaved test-then-train method from the field of data stream mining to measure the performance during the sequential learning phase. This method uses each arriving chunk for prediction to measure the performance before the same chunk is used to update and train the model [36]. This way, the model will always be evaluated on unseen data. Once the sequential learning phase is finished, i.e., when the training set is exhausted, the final prediction performance is measured on the test set when the training data are exhausted.

Three experiments were carried out to evaluate the effectiveness of the proposed approach. Below are the experiments that were carried out.

(1)Sequential learning performance. This experiment was carried out to study how well this proposed approach learned the concept from sequential data, e.g., data stream. Since this proposed approach was designed to improve the practicality of manifold regularization for data stream mining, this experiment was divided into two types of experiments:
This proposed approach versus the default RBF width parameter. This experiment compared how well this proposed approach compared with other benchmark approaches when the default width parameter was used. This experiment used γ=1number of features, which is a popular default parameter in the documentation of machine learning algorithms that relies on the RBF kernel, e.g., support vector machines (SVM).This approach versus the optimum width parameter. This experiment was carried out to compare how this proposed approach compared when the width parameter was set to an optimum value to maximize the learning performance on the benchmark algorithm. This proposed approach was effective if it achieved the same or better performance compared to other approaches, as it indicates that this proposed approach could perform well even without the RBF kernel.
(2)Effect of the number of labeled samples and classification performance. This experiment was designed to understand the effect of the size of the labeled samples on classification performance.(3)Execution time analysis. As computational consumption is a major design consideration in applications such as data stream mining, this experiment was designed to study how long this proposed approach took to execute each update compared with other approaches.

### 5.2. Hyperparameter Selection

Several hyperparameters needed to be selected in advance for both the proposed approach and the baseline approaches. For the related ELM-based approaches, which included this proposed approach, the labeled-unlabeled trade-off parameter λ except for the SSOE-ELM, which was determined dynamically, was selected from the set $\left[10^{-5},\ldots ,1\right]$, while the regularization term C was selected from the set $\left[10^{-5},\;\ldots ,\;10^{5}\right]$. The parameter that gave the fastest improvement in classification performance in the sequential learning phase was selected as the optimum parameter. This parameter selection method was preferred over other methods such as grid-search, which is impractical for large data sets. For all approaches, only three nearest neighbors (k=3) were considered to construct the affinity matrix. For the hidden layer size and the activation function of the ELM, the hyperparameter selection was more straightforward. The hidden layer size for all ELMs was set to 1000 nodes as it had been shown that it was sufficient to achieve a stable and good performance [32]. For the activation function, the rectified linear united (ReLU) was used to prevent the vanishing gradient problem, which may dampen the learning ability of the model [37]. The parameters for the benchmark approaches are summarized in Table 2.

For this approach’s hyperparameters, it did not require the width parameter γ of the RBF kernel to be determined, which was in line with the goal of this proposed approach. Additionally, to show that our proposed approach did not require any additional hyperparameter settings, we used M=3 for the number of prototypes to be paired for each piece of data for all data sets. Meanwhile, for the regularization parameter C and the trade-off parameter λ, we used the parameter C=1 and λ=0.0001 as the default parameters as they are the most frequent settings in the benchmark approaches, as shown in Table 2. Heuristically determining these parameters would be an avenue for future works. However, as shown in Table 2, C=1 and λ=0.0001 were good enough for most data sets. Therefore, they worked well as default parameter settings.

Finally, concerning the node pruning interval of the SOINN architecture to remove noisy nodes, the parameter selection was based on a computational budget rather than for classification performance. This was due to the pruning algorithm that required visiting each node to determine whether it should be removed or retained. In this study, the interval was set to 1000 samples as it allowed the ESOINN to accumulate enough nodes but not too many that it was time consuming to iterate through all the nodes. This approach was implemented using the Python programming language on a platform configured with the Intel i5 processor clocked at 2.40 GHz with 12 GB of RAM.

## 6. Results

### 6.1. Sequential Learning Evaluation

In this section, the results of sequential learning will be discussed. As mentioned in Section 5, we first investigated the performance of this proposed approach against other manifold regularization approaches with the default RBF kernel width parameter γ=1number of features. The experiment results are presented in Figure 3, which illustrates the learning progress of the proposed and baseline approaches as more data arrive. As mentioned earlier, the performance measure at each chunk was obtained using the interleaved test-then-train method. Each chunk contained 100 samples, of which only five samples were labeled. The results showed that this approach learned faster than other approaches in almost all data sets, particularly when challenges such as imbalanced classes were present.

In low-dimensional or balanced data sets, such as in the sensorless drive or magic gamma data sets, the performance of this approach was slightly better or comparable to other approaches. As illustrated by the sequential learning result in the sensorless drive data sets (Figure 3a), this approach had the same learning rate as the SOS-ELM approach but was slightly slower than ILR-ELM. Eventually, all approaches converged to 80% accuracy. Meanwhile, in the magic gamma data set (Figure 3b), this proposed approach initially learned more slowly than ILR-ELM and SOS-ELM. However, at around the 40th chunk, this approach overtook other approaches, learned faster than other approaches, and converged to a higher accuracy at around 79%.

In high-dimensional or highly imbalanced data sets, the proposed approach learned much faster and achieved better performance than other approaches. In the HAR and Crop Mapping data sets that had the highest number of features, it can be seen in Figure 3c,d that this approach learned much faster than other approaches with a fewer number of samples presented. For the HAR data set, it is obvious in Figure 3c that even using Euclidean distance, which usually fails on high-dimensional data sets, the proposed approach learned faster by having a higher accuracy during the initial phase of training at around 67%. In comparison, other approaches barely reached 30% accuracy. Similarly, in the crop mapping data set, shown in Figure 3d, this approach had a significant head start at around 80% and converged to a higher accuracy than other baseline approaches at around 98% accuracy.

For highly imbalanced data sets, represented by the KDDCup 1999 and PAMAP2 data sets, shown in Figure 3e,f, the proposed approach still learned faster than other approaches despite fewer samples presented to train the model. As evident in Figure 3e, the proposed approach learned significantly faster than other approaches by already achieving 97% accuracy as early as the 100th chunk. Towards the end of the training phase, this approach achieved almost 100% accuracy, much higher than other approaches. However, for the PAMAP2 data set, the performance difference was less significant as the class imbalance was less severe than in the KDDCup 1999 data set. This showed an apparent advantage of applying this approach to a highly imbalanced data set. As expected, and as shown in Figure 3f, this approach still learned faster on an imbalanced data set with a steeper learning curve. Despite this, all approaches converged to the same accuracy at around 70%.

We then compared this proposed approach against other manifold regularization approaches with the optimal RBF kernel width parameter, as shown in Table 2. This was to investigate if the proposed prototype regularized manifold regularization method could perform as well as other approaches when the best width parameter was selected to learn from a stream of unlabeled and with few labeled data. The result is presented in Figure 4a–f.

Based on the results, Figure 4a–f shows that this approach achieved a similar performance when an optimum width parameter was used on the baseline approaches. Using the sensorless drive data set in Figure 4a, this approach performed the same as the SOS-ELM approach but learned slightly more slowly than the ILR-ELM approach. For the magic gamma data set in Figure 4b, this approach also performed similar to SOS-ELM but was better than ILR-ELM and SSOE-ELM. However, this approach performed slightly worse than all other baseline approaches in the HAR data set in Figure 4c. The crop mapping data set in Figure 4d shows that this approach performed the same as the SOS-ELM and ILR-ELM approach but performed better than SSOE-ELM.

On the other hand, the KDDCup1999 data set in Figure 4e shows that this approach performed the same as the other data sets. Finally, the PAMAP2 data set showed an interesting characteristic where only this approach and the ILR-ELM consistently improved the model, whereas other approaches had a slight drop in performance. This might have been due to the noisy environment of the data set that harms the learning progress of these approaches.

In summary, these experiments showed that this approach could perform at least as well as other manifold regularization approaches. This shows that this approach could allow manifold regularization methods to be applied without determining the RBF kernel width parameter a priori, thus improving the practicality of manifold regularization for data stream mining.

### 6.2. Effect of Number of Labeled Samples

This section investigates the effect of the number of labeled data per chunk on classification performance. The performance measure was calculated by training the model on the training set with only the specified number of labeled data on each chunk and using the model for prediction on the test set to obtain the classification performance. Based on Figure 5, the proposed approach achieved the best performance given the fewest labeled data for each chunk.

In the sensorless drive data set and magic gamma data set, consistent with the previous experiment, the performance of this proposed approach was at par or slightly better than other approaches as there was no significant challenge such as high-dimensional or class imbalance in these data sets. As illustrated in Figure 5a, this approach did not have an advantage on any size of labeled data. Meanwhile, in Figure 5b for the magic gamma data set, this approach had a slight advantage on the fewest labeled data but diminished as more labeled data were presented.

On the other hand, for the HAR and crop mapping data sets, which were high-dimensional data sets, the difference in performance was more significant as this approach performed much better with the fewest number of labeled data per chunk. For example, in Figure 5c, this approach had a significant advantage over other approaches on all sizes of labeled data. In comparison, in Figure 5d, with a lower dimension, this proposed approach only had a significant advantage when only one labeled piece of data was used. However, this approach had a slight drop in performance in the HAR data set in Figure 5c as more labeled data were used. An explanation could be that introducing more labeled data could result in overfitting of the manifold regularization approach.

On the KDDCup 1999 data set, the proposed approach also performed better than other approaches by having a significant advantage on all sizes of labeled data, as shown in Figure 5e. However, as in the previous experiment, Figure 5f shows that the advantage diminished as the severity of class imbalance decreased, indicated by the equal performance of all approaches with more than two labeled pieces of data. Nevertheless, this approach managed to outperform other approaches, given the fewest number of labeled data.

Next, we analyzed the overall performance ranking on all data sets with a different number of labeled data per chunk to understand how this proposed approach compared against other benchmarked approaches. Table 3 shows the classification performance of all approaches with the best score highlighted. For each data set, we ranked the approaches based on their average accuracy; at the bottom of Table 3, the overall ranking of each approach is shown.

We investigated the ranking further by conducting the Friedman–Nemenyi post hoc test with a significance level of α=0.05. The result of the statistical test is illustrated by the critical distance diagram in Figure 6. The approaches that were not joined by the horizontal bar had significant performance differences, whereas those joined by the horizontal bar did not have significant performance differences. Figure 6 shows that the proposed approach did not perform significantly better than the ILR-ELM and the supervised RSLVQ approach but still managed to outrank these approaches. On the other hand, this approach significantly outranked the SOS-ELM, SSOE-ELM, and the supervised OS-ELM approach. The OS-ELM approach performed poorly despite being a fully supervised approach, which might have been due to the overfitting of the model since this approach lacked any regularization term that prevented overfitting.

### 6.3. Execution Time Analysis

This section compares the computational consumption by analyzing the execution time of each approach. Figure 7 shows a series of box plots that represent the distribution of execution time of the model update for every approach in each data set. By studying Figure 7, the proposed approach required slightly more time to update the model than other approaches.

The average time for this model to update the model was around 3.5 to 4 s, according to Figure 7a–f. On the other hand, the ILR-ELM required around 5 s to update the model. The slightly higher updating time of the ILR-ELM was due to the larger computational cost to multiply and compute the matrix inverse of the update equation as it had many more terms than the other approaches. This resulted in a long time being required to update the model for the ILR-ELM approach.

However, this proposed approach had many outliers in terms of execution in the majority of the data sets, where the execution time was significantly higher. This was an expected occurrence as most of these data sets were noisy data sets where they are collected via sensors, which caused the ESOINN to generate some noisy nodes. Hence, during the iteration where node removal occurred, many noisy nodes needed to be removed. Removing the nodes and updating the adjacency matrix can be computationally expensive; the updating period will be longer if many nodes in the interval are classified as noise. Despite this, an occasional increase in the computational consumption might be desired to protect the model from noise to ensure the integrity of the model.

## 7. Discussion

Based on the results from the experiments, via the prototype-based manifold regularization method, this approach can preserve the local neighborhood of each sample throughout learning. This can be shown by using the Euclidean distance on high-dimensional data sets such as HAR and crop mapping data sets to select the nearest neighbors for each sample. As mentioned in Section 1, due to the manifold assumption, high-dimensional data sets are locally Euclidean if they can be represented on a lower-dimensional manifold. Compared to other approaches, only the proposed approach successfully verified the manifold assumption as it can learn using low-dimensional distance measures on high-dimensional data. The significance of the proposed approach is that, instead of using higher-dimensional measures such as the RBF kernel that require the width parameter to be determined, lower-dimensional distance measures such as the Manhattan or Euclidean distance do not require any additional hyperparameter. This eliminates the requirement to determine the width parameter of the RBF kernel. As a result, no additional hyperparameter selection step is required, and the model could be deployed more quickly compared to other manifold regularization approaches.

Additionally, because the RBF kernel width parameter γ does not need to be determined, the practical implementation of this proposed approach is simpler than other manifold regularization approaches and could be deployed immediately in a data stream environment without an off-line hyperparameter search. Default hyperparameters could be used for other hyperparameters such as the regularization and label-unlabeled trade-off parameters (C=1 and λ=0.00001). As Table 2 suggests, these default hyperparameters are good enough for most data sets and will most likely work in many data stream applications.

The claim of this proposed approach’s superiority is supported by experimental results in Figure 3, Figure 4 and Figure 5, which show that this approach can learn faster and also can perform well when few pieces of labeled data are used. These results are the expected outcome of this experiment, suggesting that this approach improves the current state of the manifold regularization approaches in terms of practicality.

Experimental results also suggest that this approach performs well on noisy data sets such as the KDDCup 1999 and PAMAP2, which are data sets collected via sensors and may be noisy. The current approaches for manifold regularization have no mechanism to guard them against noisy data. By using the ESOINN that can filter out noisy data and use it to store the prototypes, this approach can protect the learning algorithm from learning from noisy data and could boost learning performance. Some possible observational error sources can be labeling errors from these handcrafted data sets used in the experiments that can influence the measurement of the performance measures used in the experiments.

However, protection against noisy data comes at the cost of an increase in execution time to update the model, particularly in noisy data. As the ESOINN needs to iterate through all nodes to eliminate noise-generated nodes, during intervals where many noisy nodes are being created, it will require a longer time to remove noisy nodes. To reduce the learning time, a longer interval may be used to only remove noisy nodes occasionally; therefore, the algorithm does not have to go through all the nodes many times during the learning phase.

## 8. Conclusions

This paper proposed a prototype-based manifold regularization technique to ensure that the manifold regularization term is constructed using the local neighborhood for each piece of data. In this approach, we modified the ESOINN prototype learning algorithm and combined it with the SOS-ELM manifold regularization technique but only used the prototypes from the ESOINN to construct the manifold regularization term. We compared this approach with other approaches when the default RBF width parameter was used and when an optimum RBF width parameter was used for the benchmark approaches. Based on the experiments, the proposed approach achieved a faster learning rate and higher classification performance on imbalanced and high-dimensional data sets compared to other benchmark approaches when the default RBF width parameter was used. The proposed approach also performed as well as other approaches when the optimum RBF kernel width parameter was used for the benchmark approaches, indicating that this approach could perform well even without a hyperparameter search. A study to compare the effect of the size of the labeled samples for each approach showed that this approach consistently improved its performance the most when more labeled data were added to each chunk. According to the hypothesis test, this approach significantly outranks some current manifold regularization approaches. However, this comes at the cost of higher computational consumption to train the ESOINN. In future work, we suggest designing an approach that could determine all the hyperparameters rather than relying on the default hyperparameters. Ideally, this approach should be able to determine all the hyperparameters online to suit data streaming applications.

## Figures and Tables

**Figure 1 sensors-22-03113-f001:**
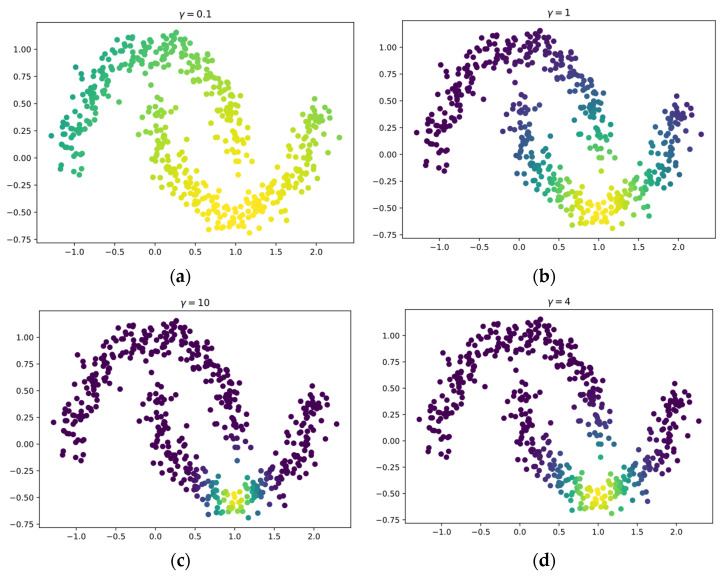
Illustration of the importance of the width parameter γ of the RBF kernel from semi-supervised learning; (**a**) shows a poor selection of the width parameter γ=0.1  that misleads the model into perceiving that all data belong to the same class; (**b**) shows a better selection of the width parameter γ=1 as most of the mass is concentrated on data in the same class but touches many of the data in the opposite class; (**c**) also shows a poor selection of the width parameter γ=10 as it only indicates that only the most similar data from the center belong to the same class; (**d**) shows the best width parameter γ=4 among the other four as it specifies a wide enough region of data that belong to the same class.

**Figure 2 sensors-22-03113-f002:**
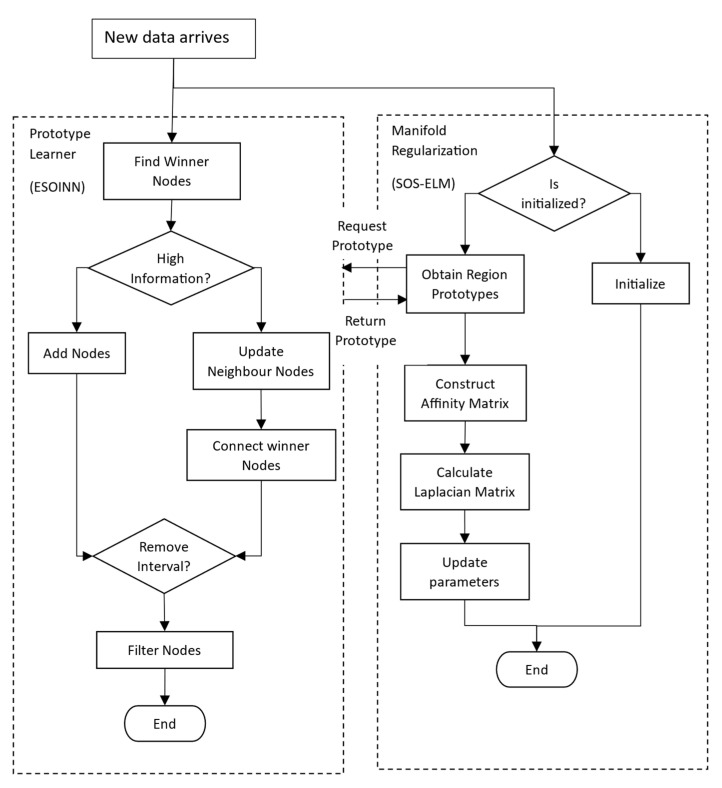
Overview of the proposed approach.

**Figure 3 sensors-22-03113-f003:**
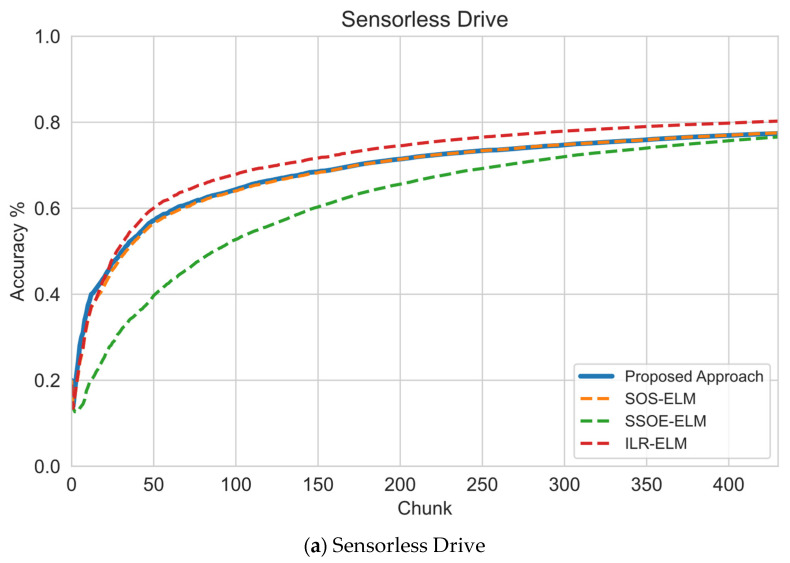
Sequential learning evaluation of this approach versus default RBF kernel hyperparameters for other approaches: (**a**) sensorless drive data set showed this approach performed comparably to other approaches; (**b**) magic gamma data set showed this approach learned faster than other approaches; (**c**) HAR showed this approach significantly learned faster than other approaches as it was a high-dimensional data set; (**d**) crop mapping, which was also a high-dimensional data set, also suggested that this approach can perform well on a high-dimensional data set; (**e**) on imbalanced data set, the KDDCup1999 data set showed that this approach can learn faster than other approaches; (**f**) on PAMAP2 data set, this approach learned faster than other approaches but converged to the same accuracy.

**Figure 4 sensors-22-03113-f004:**
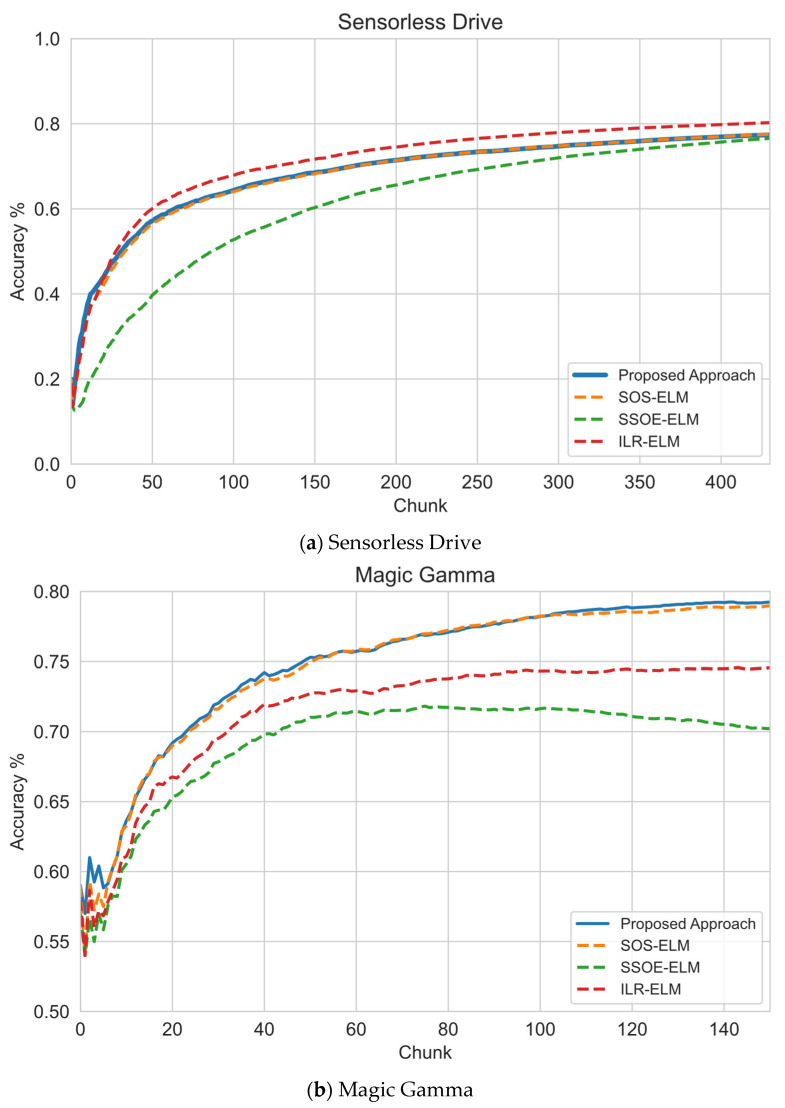
Sequential learning evaluation of this approach versus optimal RBF kernel hyperparameters for other approaches: (**a**) on sensorless drive data set, this approach performed second best but was comparable to the SOS-ELM approach; (**b**) on magic gamma data set, this approach’s performance was comparable to the best approach, ILR-ELM; (**c**) on HAR data set, this approach’s performance was slightly worse than other approaches; (**d**) on crop mapping data set, this approach performed close to other approaches except SSOE-ELM; (**e**) on KDDCup1999 data set, all approaches had a similar learning performance; (**f**) on PAMAP2 data set, this approach had a more stable learning performance compared to other approaches when exposed to a noisy data set.

**Figure 5 sensors-22-03113-f005:**
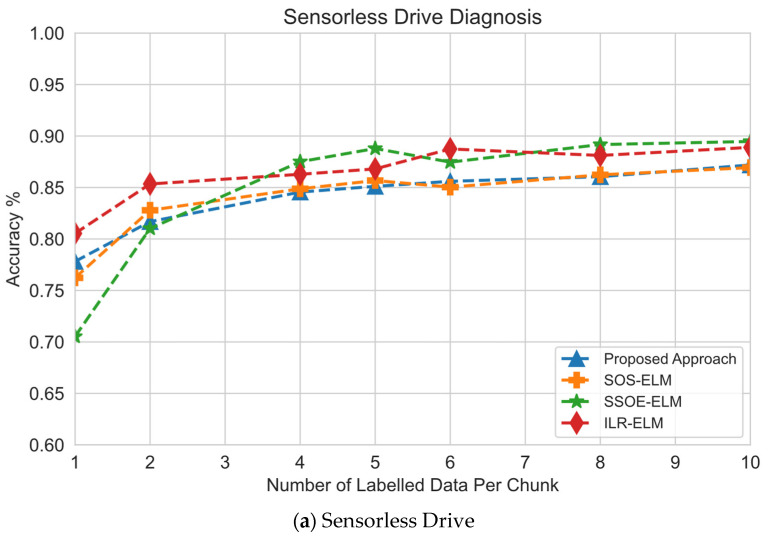
Effect of the size of labeled data on classification performance: (**a**) on sensorless drive data set, this approach performed second best when the fewest number of labeled data was used but no significant improvement to other approaches; (**b**) on magic gamma data set, this approach performed the best regardless of the number of labeled data used; (**c**) on HAR data set, this approach significantly outperformed other baseline approaches on all labeled data sizes but decreased slightly, possibly due to overfitting; (**d**) on the crop mapping data set, this approach performed the best on the fewest labeled data sizes but performed comparably to other approaches as the size of labeled data increased; (**e**) on KDDCup 1999 data set, this approach performed the best when the fewest labeled data were used but the advantage diminished as the size of labeled data increased; (**f**) on PAMAP2 data set, this approach performed the best on the fewest labeled data sizes but performed comparably to other approaches as the size of labeled data increased.

**Figure 6 sensors-22-03113-f006:**
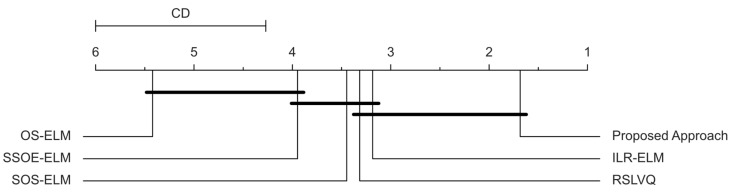
Critical distance (CD) diagram for the Friedman–Nemenyi post hoc test for comparison of this approach against other approaches.

**Figure 7 sensors-22-03113-f007:**
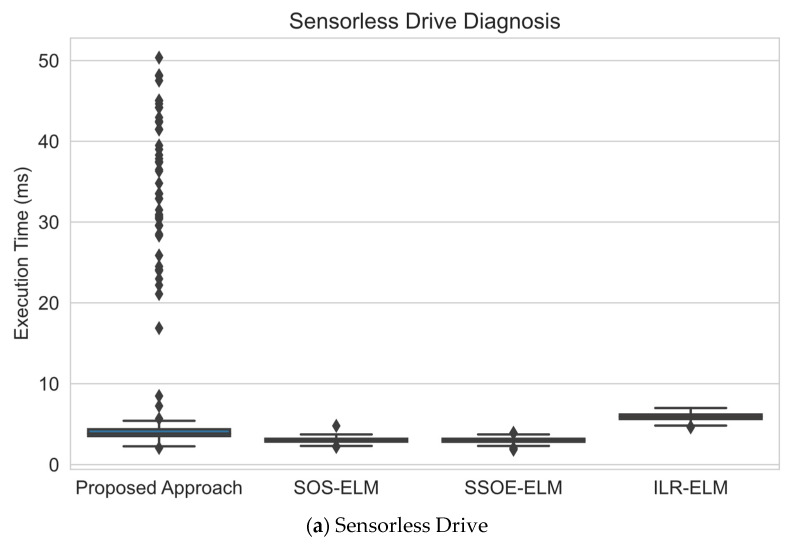
Execution time distribution comparison among approaches in milliseconds (ms): (**a**) on sensorless drive data set, this approach required less time to update the model compared to ILR-ELM but had many outliers to remove the noise; (**b**) this approach required less time than ILR-ELM to update the model on the magic gamma data set but also had outliers in execution time; (**c**) on the HAR data set, the execution time required was also less than only ILR-ELM’s but its mass was mostly located towards a lower execution time (right skewed); (**d**) on the crop mapping data set, this approach took the longest time to update but its execution time was mostly towards a lower execution time (right skewed); (**e**) on the KDDCup 1999 data set, this approach took less time on average than ILR-ELM to update the model but it also had many outliers in execution time; (**f**) on PAMAP2 data set, despite having an execution time comparable to other approaches with less execution time, this approach also had many outliers in the execution time.

**Table 1 sensors-22-03113-t001:** Benchmark data sets.

Data Set	#Samples	#Features	#Classes	#Training Samples	#Test Samples	#Training Steps	#Labeled Samples	Data Characteristics
Sensorless Drive	58,509	49	11	46,807	11,701	469	2340	High-Dimensional
Magic Gamma	19,020	11	11	15,216	3804	152	760	Noisy/Class Imbalance
Human Activity Recognition (HAR)	10,299	561	6	5881	4418	58	290	Noisy/High-Dimensional
Crop Mapping	325,834	175	7	60,000	65,167	600	3000	High-Dimensional
KDDCup 1999	494,021	42	12	50,000	98,805	500	2500	Class Imbalance
Physical Activity Monitoring (PAMAP2)	1,942,872	52	12	50,000	388,575	500	2500	Noisy/High-Dimensional

**Table 2 sensors-22-03113-t002:** Hyperparameter settings for manifold regularization approaches.

Data Set	γ	λ	C
Crop Mapping	0.001	0.00001	1
HAR	10	$10\times 10^{-5}$	0.001
KDDCup1999	20	0.00001	1
Magic Gamma	1	0.00001	1
PAMAP2	10	0.00001	1
Sensorless Drive	0.01	0.00001	1

**Table 3 sensors-22-03113-t003:** Performance comparison of this approach compared with other related semi-supervised learning techniques and supervised learning approaches.

Data Set	Number of Labels	This Proposed Approach	SOS-ELM	SSOE-ELM	ILR-ELM	OS-ELM	RSLVQ
Crop Mapping	1	**96.06**	81.03	80.16	86.63	83.29	88.43
	5	**98.31**	98.2	98.16	98.2	82.54	88.43
	10	**98.56**	98.58	98.45	98.58	83.88	88.43
**Average Accuracy**		**97.64**	92.60	92.26	94.47	83.24	88.43
**Average Ranking**		1	3	4	2	6	5
HAR	1	73.95	30.99	31.33	31.26	70.96	**81.37**
	5	**87.01**	51.22	49.12	50.86	70.94	81.37
	10	**84.9**	66.39	66.18	68.02	71	81.37
**Average Accuracy**		**81.95**	49.53	48.88	50.05	70.97	81.37
**Average Ranking**		1	5	6	4	3	2
KDDCup 1999	1	**99.05**	91.64	90.4	68.02	18.3	98.1
	5	**99.59**	98.37	96.92	98.64	87.8	98.1
	10	**99.72**	99.34	98.61	99.45	78.35	98.1
**Average Accuracy**		**99.45**	96.45	95.31	88.70	61.48	98.10
**Average Ranking**		1	3	4	5	6	2
Magic Gamma	1	**80.55**	70.37	75.74	70.87	69.14	79.34
	5	**81.73**	80.63	67.59	77.08	66.93	79.34
	10	**83.99**	82.6	72.11	78.89	69.77	79.34
**Average Accuracy**		**82.09**	77.87	71.81	75.61	68.61	79.34
**Average Ranking**		1	3	5	4	6	2
PAMAP2	1	**68.35**	62.81	62.57	62.41	4.2	62.54
	5	83.62	84.86	84.87	**83.97**	30.26	62.54
	10	**89.1**	88.82	88.91	88.53	29.52	62.54
**Average Accuracy**		**80.36**	78.83	78.78	78.30	21.33	62.54
**Average Ranking**		1	2	3	4	6	5
Sensorless Drive	1	77.82	76.23	70.53	**80.54**	48.46	81.27
	5	85.11	85.68	**88.79**	86.81	46.1	81.27
	10	87.19	86.93	**89.47**	88.9	43.16	81.27
**Average Accuracy**		**83.37**	82.95	82.93	85.42	45.91	81.27
**Average Ranking**		2	3	4	1	6	5
**Overall Average** **Accuracy**		**87.48**	79.7	78.33	78.76	58.59	81.84
**Overall Ranking**		1.72	3.36	3.83	3.31	5.39	3.39

## Data Availability

For more information on the data sets used in the experiments, please visit: (1) Sensorless Drive Diagnosis data set, 2015. UCI Machine Learning Repository: Data set for Sensorless Drive Diagnosis Data Set; (2) MAGIC Gamma Telescope data set, 2007. UCI Machine Learning Repository: MAGIC Gamma Telescope Data Set; (3) Human Activity Recognition (HAR) data set, 2012. UCI Machine Learning Repository: Human Activity Recognition Using Smartphones’ Data Set; (4) Crop Mapping Using Fused Optical Radar data set, 2020. UCI Machine Learning Repository: Crop mapping using fused optical-radar data set; (5) Knowledge Discovery and Data Mining Competition (KDDCup) data set, 1999. KDD Cup 1999 Data (uci.edu); and (6) Physical Activity Monitoring (PAMAP2) data set, 2012. UCI Machine Learning Repository: PAMAP2 Physical Activity Monitoring Data Set.

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
