# Peer review of "Prototype Regularized Manifold Regularization Technique for Semi-Supervised Online Extreme Learning Machine"

_sensors, 2022, doi:10.3390/s22093113_

Round 1
Reviewer 1 Report
The topic Prototype Regularized Manifold Regularization Technique for Semi-Supervised Online Extreme Learning Machine is potentially interesting, however, there are some issues that should be addressed by the authors: The Introduction" sections can be made much more impressive by highlighting your contributions. The contribution of the study should be explained simply and clearly. The authors should further enlarge the Introduction with current work about artificial intelligence techniques and monitoring platforms to improve the research background, for example: Online semi-supervised learning with learning vector quantization; Effective multi-sensor data fusion for chatter detection in milling process; Effective feature selection with fuzzy entropy and similarity classifier for chatter vibration diagnosis.
Clarify the practical implementation of the proposed strategy
Clarify how you adjust the parameters of the proposed approach
Increase the resolution of Figures 3, 4, 5, 7
Conclusion section should be rearranged. According to the topic of the paper, the authors may propose some interesting problems as future work in the conclusion.
This study may be proposed for publication if it is addressed in the specified problems.
Reviewer 2 Report
The authors face a very interesting aspect: how to proceed for sequential learning from partially labelled data, when manifold regularization technique does not work for RBF kernel. They propose to remove the RBF kernel from the manifold regularization and combine the manifold regularization with a prototype learning method, which uses a finite set of prototypes to approximate the entire dataset.
According to their results, the proposal can learn faster and achieve a higher classification performance, compared to benchmark datasets.
The paper is well structured, it clearly states the approach, the novelty and the contributions. The methodology is detailed and the proposal is compared to other approaches using several benchmark datasets. The results show promising results.
In my opinion, the paper can be published in its current state.
Reviewer 3 Report
Dear Authors,
The content of your article fits perfectly into the scope of a Sensors journal. There is no doubt that the article deserves to be published. One of the reasons for this is that the key research topics involves the machine learning problem of processing or predicting from sequential data from different sources, such as sensors and mobile devices, for example in Internet of Things applications, which is currently very important due to the generation of large amounts of data.
The authors proposed a prototype-based manifold regularization method. It is relevant and interesting.
The above-mentioned goal was based on the 29 publications analysed in the three initial sections of the article.
Experimental results are given and explained. The authors proved that the technique they have developed has faster learning speed and higher classification performance on imbalanced and high dimensional datasets.
The paper contains some new data.
The paper is presented in logical way and overall written well.
The text is clear and easy to read.
The conclusion is consistent with the evidence and arguments presented and addresses the main question asked. The article makes reference to limitations and future research.
Comments and Suggestions for Authors
- It would be best to clearly identify which of the previous works by all Authors constitute the foundation of the work presented in this article.
- The study considered the use of the KDDCup 1999 and PAMAP2 datasets. What other alternatives were there?
- Are the correlations of the experimental results obtained, which are shown in Figures 3-5, optimal? What are the observational errors?
- The following typos are noticed in the article: please make the content of Figure 7 more readable (on pp. 21-22).
Round 2
Reviewer 1 Report
The authors handled the comments